# Differences in Stress, Stress-Coping Behavior, and Quality of Life Based on the Performance of Korean Ladies Professional Golf Association Tour Players

**DOI:** 10.3390/ijerph19116623

**Published:** 2022-05-29

**Authors:** Soon-Young Kim, Chulhwan Choi

**Affiliations:** Department of Physical Education, Gachon University, 1342 Seongnam-daero, Sujeong-gu, Seongnam-si 13120, Korea; klpga0166@gachon.ac.kr

**Keywords:** stress, stress coping behavior, quality of life, performance, KLPGA

## Abstract

Maintaining world-class performance, irrespective of the influence of various psychological factors, is the most important task for professional athletes. By recognizing and coping with profession-related stress, athletes can improve their performance and maintain their quality of life as a professional. This study compared and analyzed the stress, stress-coping behavior, and quality of life of world-class athletes based on their objective performance. Data were collected from 234 professional golfers active on the Korean Ladies Professional Golf Association Tour and Dream Tour. Using performance as an independent variable, one-way multivariate variance analysis was performed for comparative analysis. The results indicated that professional tour golf players showed statistically significant differences in (a) stress from fellow players, (b) performance-related stress, and (c) passive stress-coping behavioral factors. Groups with higher levels of performance experienced more stress than their counterparts and coped with stress through a more passive attitude. Importantly, efforts to improve performance under fierce competition and lead a better life are essential for maintaining psychological stability.

## 1. Introduction

The size and popularity of women’s professional golf in South Korea have been growing since the late 1990s [1]. An important driver of this growth has been the successful performance of Korean professional golfers overseas for more than 20 years. Since 1998, Korean players have won 15 Rookie of the Year awards from the Ladies Professional Golf Association (LPGA), the leading golf tour for female players. Most players have refined their skills on the Korean Ladies Professional Golf Association (KLPGA) Tour [2]. These achievements have been possible because golfers present their best performance as professional sportspersons. Ultimately, owing to the efforts of each player, the fierce competition between players, and support from the KLPGA, Korea has been steadily producing some of the world’s most competitive players for more than 20 years.

Specifically, this growth in Korean women’s professional golf began with the ascent of Pak Seri, who was also part of the golfing Hall of Fame. Seri’s 1998 LPGA US Women’s Open Golf Championship performance is widely regarded as a message of hope to Koreans amid the economic crisis of the late 1990s [3]. Previously, golf was not a popular sport among Koreans because of its high participation cost and perception as a leisure sport enjoyed by the elite. However, golf began gaining recognition after Seri won a series of international competitions. The popularity of the sport among Koreans has witnessed a remarkable upswing that continues today. Most importantly, Seri’s success not only attracted public participation, but also gave many young players the confidence that they could also become world-class players. Soon, a series of young golfers, called “Seri Kids,” who wished to achieve glory as professional sports players, appeared on the global stage. For example, all of the top 10 women professional golfers began their careers as youth players. Today, these Seri Kids have grown into icons themselves, positively influencing other young players [1].

At her retirement press conference, Seri stated, “I’ve been a golfer since childhood and tried my best to win the competition against top players. However, I lacked preparation for a normal life other than my life as a golfer, and it was always mentally difficult” [4]. The interview resonated greatly with many young players who hoped to become as legendary as Seri Pak. The statement caused professional golfers and youth players, who spend substantial time and effort to maintain outstanding professional performance while making considerable personal sacrifices, to reflect on their lives. The players became more aware of the importance of balancing their daily life with their competitive life as professional golfers, an aspect that is easily overlooked when facing fierce competition. Importantly, factors such as stress caused by extreme tension, coping behaviors to relieve stress, and quality of life are now being recognized as essential factors in maintaining the psychological health of professional athletes. Accordingly, many Korean women’s professional golf players have consistently attempted to strike a balance between their personal and professional lives while also steadily trying to improve their professional performance in a mentally stable state [5]. Today, the need to manage mental health and quality of life has become an important precursor to professional success.

Stress has become an inherent part of our daily lives. The types of stress people experience and how they deal with it are important factors in determining their quality of life in various situations [6]. Research has demonstrated that stress is an important factor in people’s physical, mental, and social health. Stress can be defined as a physiological and non-specific reaction to external or internal demands [7]. Alternately, stress arises from the discovery of differences between individuals’ actual and desired states [8]; various mental factors that act on this gap cause stress. Most studies have analyzed the relationship between stress and various major factors rather than analyzing a single factor [9]. They provide extensive evidence on the role of stress in people’s overall health. Notably, stress causes tension, irritation, and fatigue, which are considered significant negative factors in elite sports. In the context of golf, stress-related factors can be divided into sub-factors such as stress from other players’ performance, financial stress, and stress from external factors [10]. Studies related to the stress of professional golfers at the international level have been on long journeys during tours [11], training programs [12], and real-time stress during competitions [13].

The concept of stress-coping behavior has emerged because of the importance of stress-related research. Stress is prevalent in daily life, and one cannot practically prevent it as it arises from the gap between desire and reality. Therefore, individuals’ stress-coping behaviors are important, considering the potential long-term benefits of managing stress more effectively. Coping behaviors can have positive outcomes by directly or indirectly eliminating or changing the determinants of stress [14]. A coping strategy can be defined as a dynamic cognitive and behavioral effort to manage external or internal needs [15], as an effort to cognitively control the stress caused by the difference between perceived and desired states. Furthermore, designing appropriate measures is difficult because one may not be able to predict how and why stress may manifest in a specific place or situation [16]. Stress-coping behavior starts with recognizing and confirming the degree of stress in situations. This is followed by identifying an individual’s resources for overcoming stress [17]. Through this cognitive process, stress-coping behavior can be divided into two actions: (a) problem-solving-oriented active coping behaviors [18] and (b) passive coping behaviors that focus on minimizing the negative effects of stress through avoidance or neglect [19]. The prevailing opinion is that active coping behaviors produce positive results in terms of eliminating the cause of stress [20,21].

The ultimate purpose of developing stress-coping behaviors is to improve the quality of life. Quality of life is a relative rather than an absolute concept that varies from person to person or situation to situation [22]. Many researchers have sought to investigate (a) whether the concept of quality of life is adequately comprehensive and (b) whether various factors affect the quality of life [23]. Studies have confirmed that quality of life is a comprehensive and subjective concept in line with the concept of happiness [24]. With such a broad and ambiguous conception, quality of life also provides a positive direction, because most people want to positively evaluate their lives [25,26]. Other studies have applied factors such as job saturation; work-to-leisure connections [27]; and physical, material, social, emotional, and psychological wellbeing to the quality of life [28]. This study attempts to determine how world-class professional athletes can pursue a high quality of life, while giving their best performance, by analyzing the stress and stress-coping factors experienced by professional golf tour players.

Accordingly, this study aims to compare and analyze the stress, stress-coping behavior, and quality of life of Korean women’s professional golf tour players based on objective performance data. The type of tour was applied as a criterion for classifying objective performance: KLPGA tour (major tour, official KLPGA tour) and Dream Tour (minor tour, development tour under KLPGA). This is because participation in each tour category, determined by the players’ cumulative results, can serve as the most objective indicator of professional golfers’ performance. More importantly, this study offers an opportunity to analyze players’ attempts to balance their lives as successful athletes and their daily lives as individuals by analyzing (a) the types of stress they face, (b) how they manage the stress, and (c) their quality of life as professional golfers. This study is particularly relevant because all of the professional golf athletes who participated in this study were active players on the professional golf tour in 2021. The data on these active players can be applied to understand the psychological states and quality of life of many professional athletes.

The study sought to answer the following research question: Do active women’s professional golf tour players exhibit meaningful differences in terms of stress, stress-coping behavior, and quality of life depending on their performance? Specifically, the purpose of this study is to compare and analyze the differences in stress, stress coping, and quality of life based on the type of professional golf tour (i.e., objective performance).

## 2. Materials and Methods

### 2.1. Participants and Data Collection

This study surveyed 128 full-time KLPGA members who participated in the KLPGA 2021 High1 Resort Women’s Open, held at the High1 Country Club for four days, from 19 to 22 August 2021. In addition, a survey was conducted with 125 participants in the 2021 Tolvist and Phoenix Country Club Dream Tour, a development tour, from 23 to 24 August 2021. The purposive sampling method was used to obtain data from golf tour players (only KLPGA and Dream Tour players were eligible to participate in this study). Additional exclusion criteria were not applied. This study refrained from collecting respondents’ sensitive personal information, and participation in the survey was voluntary. In addition, the professional golf tour players were fully aware of their rights as survey respondents, including the fact that they could stop at any time while completing the questionnaire.

Of the 253 questionnaires distributed, 249 were collected (125 and 124 were KLPGA and Dream Tour players, respectively). After excluding 15 incomplete surveys (14 and 1 from KLPGA and Dream Tour players, respectively), 234 were finally used. Based on the G-power program 3.1.9.7 (statistical power = 0.0625 and effect size = 0.90) requiring a minimum of 192 samples, the 234 surveys in this data met the statistical standard. First, the professional golfers who participated in this study responded to demographic and sociological questions, including those regarding their age, golf experience, and winning experience. Next, for segmentation, an essential requirement in comparative research, all participants answered a question about the type of golf tour in which they were currently participating, which was an important independent variable. Based on their responses, the survey respondents were divided into two groups (Group 1 = KLPGA Tour players; Group 2 = Dream Tour players). Finally, the participants responded to questions related to the three factors (stress, stress coping, and quality of life) adopted as the dependent variables. The descriptive statistics are reported in Table 1.

### 2.2. Instrument

To investigate the stress experienced by professional golf tour players based on the type of golf tour, a revised version of Kim [10] (four sub-factors; 17 items) and Kim’s [29] instrument (two sub-factors; 38 items) measuring stress and exercise burnout in golfers (amateurs and athletes) was used. Next, to examine the stress-coping behaviors of professional golf tour players, an instrument by Cho and Kim [30], who investigated stress-coping behaviors during the COVID-19 pandemic, was utilized. Finally, to measure the quality of life of professional golf tour players, the study utilized the instrument applied by Jung [31], who investigated the relationship between job satisfaction and quality of life.

Before collecting data, this study conducted a series of pretests (i.e., a panel of expert and field tests) to ensure the validity, reliability, and readability of the instruments. A panel of experts, including four faculty members and two researchers specializing in sports psychology and sports management, checked all instruments to match the purpose of this study. Based on the feedback from the experts, the instruments were selected and applied after minor revisions. Additionally, a field test with graduate students in physical education was implemented to ensure the readability of the instruments. Depending on the feedback from the graduate students, the survey questionnaires were minorly revised (e.g., wording and question order).

Consequently, this study confirmed an instrument consisting of stress (four sub-factors with 13 items), stress coping behavior (two sub-factors with eight items), and quality of life (a single factor with four items). Detailed survey questions on stress and stress-coping behavior are reported in Table 2 and Table 3. The factor of quality of life included four items, as follows: “I have achieved what I wanted so far,” “Most of what I do is rewarding,” “I do not regret my past life,” and “I am satisfied with most of my current life.” All survey questionnaires used a five-point Likert-type scale, ranging from “strongly disagree” (1 point) to “strongly agree” (5 points).

### 2.3. Data Analysis

Data analysis was performed using SPSS version 23.0. First, the analysis reported the descriptive statistics, including the sociodemographic information of all survey respondents. Next, to ensure the validity of the data collected, two exploratory factor analyses (EFAs) were performed for two of the three dependent variables (i.e., stress and stress coping). Cronbach’s alpha was used to verify the data reliability. Finally, a multivariate analysis of variance (MANOVA) was performed to determine the differences in dependent variables between the two groups (type of professional golf tour).

## 3. Results

### 3.1. Validity and Reliability

The two EFAs using principal component analysis (PCA) were separately conducted for each dependent variable: (a) stress (fellow player (four items), financial (three items), family and coaches (three items), and performance (three items)), and (b) stress coping (active stress coping (four items) and passive stress coping (four items)). Quality of life was excluded from this factor analysis because it is a single-scale factor.

First, regarding the factor structure of stress factor, the Kaiser–Meyer–Olkin (KMO) measure confirmed sample adequacy (0.851) [32]. In addition, Bartlett’s test of sphericity showed statistical significance (*χ^2^* = 2642.186, *df* = 78, *p* < 0.01). The remaining four factors, explaining 83.638% of the total variance, had eigenvalues greater than one and factor structure coefficients greater than 0.40. Regarding the internal consistency, the following factors showed acceptable (greater than 0.70) Cronbach’s alpha coefficients for internal consistency: stress from fellow players (*α* = 0.897), financial stress (*α* = 0.943), family/coach stress (*α* = 0.899), and performance stress (*α* = 0.898) [33] (Table 2).

Next, regarding the factor structure of coping with stress, the KMO measure revealed sample adequacy (0.749) [32]. In addition, Bartlett’s test of sphericity revealed statistical significance (*χ^2^* = 1057.104, *df* = 28, *p* <0.01). This analysis retained two factors, accounting for 71.486% of the total variance, with eigenvalues greater than one and factor structure coefficients greater than 0.40. Furthermore, the factors indicated acceptable Cronbach’s alpha coefficients for internal consistency: active stress coping (*α* = 0.852) and passive stress coping (*α* = 0.853) [33] (Table 3). Finally, the factors of quality of life (*α* = 0.873) and performance (*α* = 0.861), which were excluded from this factor analysis as a single factor, also showed acceptable reliability.

### 3.2. Multivariate Analysis of Variance

A MANOVA was performed to determine the differences in stress, stress coping, and quality of life and their impact on life satisfaction. Furthermore, the homogeneity of covariance was tested (Box’s *M* = 38.912, *F* = 1.345, *p* >0.001). Statistically significant differences were observed between the two groups (*F* (7, 226) = 5.055, *p* = 0.00, Wilks’ lambda = 0.864, partial *η*^2^ = 0.135) for (a) stress from fellow players, (b) performance-related stress, and (c) passive stress coping. However, no significant differences were found for (a) financial stress, (b) stress from family and coaches, (c) active stress coping, and (d) quality of life. The detailed information from the analysis is reported in Table 4.

The mean scores and standard deviations of the dependent variables for each group are reported in Table 5.

## 4. Discussion

This study compared and analyzed the stress, stress-coping behavior, and quality of life of 234 active world-class athletes based on their objective performance. This research shows that professional athletes experience difficulties in maintaining their best performance while managing psychological stability and quality of life. Moreover, this study offers an important dataset on more than 200 active players, which can be meaningfully utilized in future research regardless of the sport.

Sustaining top-notch global performance is an ability that only a few outstanding professional athletes possess. This is because players need not only physical ability, but also to deal with and resolve psychological factors that affect performance. In addition, they must maintain a good quality of life as individuals, even as they continue to grow as professional athletes. Thus, all of the aforementioned factors (i.e., stress, stress-coping behavior, and quality of life) may affect professional players’ performance and their daily lives. This study compared and analyzed the differences in athletes’ stress, stress-coping behavior, and quality of life according to their objective performance using survey data collected from professional tour golfers in the KLPGA.

First, stress from fellow players showed a statistically significant difference. Specifically, this stress is caused by the attitudes of fellow players in the game, or participants’ relationships with them. In general, professional players whose performance is ranked and evaluated through competitions face fierce professional competition. Based on the results of previous studies, stress is a dominant factor for elite athletes because it generally changes their psychological state [10]. In particular, stress tends to have serious consequences, such as suspension of exercise [34] and loss of interest in exercise [35]. In this study, stress from fellow players was higher in the high-performance group (Group 1). This may be because of factors such as higher levels of competition, pressure to maintain current performance, and confidence as an athlete [10]. In the case of golf, the stress from fellow players is inevitably more important, because golfing championships are not an absolute record-measuring event, but rather a relative event in which performance is determined by showing better results compared to the opponent. The results indicate that, to maintain objective performance, it is important to consider how individuals relieve stress under fierce competition to perform better.

Next, we also analyzed the stress from performance, which was identified as the highest stressor among all study participants. Even if current performance is excluded, professional players are always under extreme stress to boost their performance. Moreover, players who already maintain a relatively high level of performance exhibited higher stress. The effort and enthusiasm of elite athletes to improve their performance, regardless of the event, are enormous. However, professional players’ experience of stress in trying to maintain their best performance leads to a sense of shame, failure, and discouragement, which they should be careful about as it can eventually cause exhaustion [36]. This psychological state may be a response to chronic rather than temporary stress and the result of a process in which they are under long-term stress [37]. Stress from unsatisfactory performance can lead to emotional exhaustion during practice and games [38]. As shown here, even world-class players who already had a high level of performance were relatively more stressed about their performance, and this trend continued even when the performance level increased. This psychological instability can be improved through social support or mental training, which can be supplemented with support from fans or continuous consultation with psychotherapists [39].

Finally, higher results were found in the group with a relatively high level of objective performance in passive-coping factors, which are sub-factors of stress coping factors. This indicates that top players on the KLPGA official tour tend to try to forget emotional pain through passive actions such as avoidance and neglect, rather than trying to change the problem itself by focusing on positive efforts or actions. Most KLPGA official tour players are highly popular with the public and continue to communicate with their fans through social media. These seem to be passive rather than active responses. Hence, players adopt passive stress-coping behavior because maintaining a positive image in the media as a highly popular professional golf tour athlete is advantageous for sponsorships and popularity. However, studies have shown that active coping behavior results in effective stress reduction [40]; this is because a stable mental state allows an individual to perform better.

Despite meaningful research achievements, this study has some limitations. First, the comparative analysis has some general limitations. Specifically, this study did not analyze the relationships among objective performance, stress, stress-coping behavior, and quality of life. This is because the relationships or influences between factors have been analyzed in many previous studies. Nevertheless, this study may have yielded different results than previous studies. Therefore, future studies should consider extensively analyzing these relationships using the data from this study.

Second, the survey research methodology imposes some limitations. Cross-sectional research can be an effective research method to analyze the current situation based on the opinions of many participants. However, there may be a limit to collecting the complex and subjective opinions of individuals. This study did not examine the psychological status of tour players, who may have different stress and coping behaviors. As previously stated, analyzing relative and subjective concepts may be insufficient in the case of quality of life. Therefore, future studies should analyze the psychological state of professional athletes from a new perspective by applying a qualitative research design.

Finally, all survey respondents were current professional athletes. While this may increase the importance of the results of this study, this study realizes that surveys of retired players can also provide meaningful data. Accordingly, research is currently underway to analyze the psychological status of professional golf tour athletes, including retired ones. This research approach can be used to derive results that can be applied not only to academic contributions, but also to all professional athletes.

## 5. Conclusions

The research purpose was to confirm whether the difference in the objective performance of professional athletes also differs based on their psychological factors. The results indicated various differences between the two groups that objectively differed in performance and provided a meaningful interpretation. For success, professional athletes need a cautious approach—not only physically or technically, but also psychologically. This study was an opportunity to reaffirm how important the psychological aspect is for professional athletes who always have to maintain or improve their best performance. It was also confirmed that life as a professional athlete is as important as life as a human being. Eventually, it is in line with the recent emergence of the importance of work–life balance. An approach to life with the proper balance can be applied to many people, regardless of their varying situations.

## Figures and Tables

**Table 1 ijerph-19-06623-t001:** Descriptive statistics of the sample.

		Group 1	Group 2
		KLPGA Tour Players	Dream Tour Players
Age	10 s	1 (0.9%)	10 (8.1%)
	20 s	93 (83.8%)	106 (86.2%)
	30 s	16 (14.4%)	5 (4.1%)
	Over 40	1 (0.9%)	2 (1.6%)
Career as a golf athlete (years)	1–5	2 (1.8%)	2 (1.6%)
	6–10	20 (18.0%)	53 (43.1%)
	11–15	66 (59.5%)	53 (43.1%)
	Over 16	23 (20.7%)	15 (12.2%)
Professional tour win	None	11 (9.9%)	70 (56.9%)
	Once	43 (38.7%)	34 (27.6%)
	Twice	29 (26.1%)	9 (7.3%)
	Three times	13 (11.7%)	9 (7.3%)
	Four times	3 (2.7%)	-
	Over five times	12 (10.8%)	1 (0.8%)
Total		111 (100%)	123 (100%)

*Source*: Authors’ study.

**Table 2 ijerph-19-06623-t002:** Factor matrix structure for stress.

Items	1	2	3	4
I am stressed out by my fellow players’ unsportsperson-like behavior	**0.930**	0.08	0.149	0.141
I am stressed out by my fellow players’ rude behavior	**0.92** **2**	0.124	0.149	0.217
I get stressed when my fellow players are not considerate	**0.859**	0.167	0.158	0.258
I am stressed out about my relationship with my fellow players	**0.500**	0.254	0.339	0.192
I am under economic stress during my career	0.145	**0.911**	0.209	0.161
I am stressed out because my career is too costly	0.149	**0.900**	0.107	0.221
I am stressed out by financial burdens during my career	0.146	**0.884**	0.233	0.161
Parents’ expectations from me are stressful	0.177	0.114	**0.882**	0.185
Coaches’ expectations from me are stressful	0.15	0.222	**0.862**	0.188
My family’s expectations from me are stressful	0.187	0.179	**0.853**	0.036
I get stressed when I cannot play as I want during the game	0.144	0.227	0.14	**0.856**
I get stressed when I feel limited in my performance	0.261	0.214	0.154	**0.853**
I get stressed when my golf skills do not improve	0.253	0.111	0.126	**0.850**
Eigenvalues	6.185	1.835	1.586	1.267
Variance (%)	45.579	14.115	12.201	9.744
Cronbach’s alpha	0.897	0.943	0.899	0.898

Note: 1 = fellow player, 2 = financial, 3 = family and coaches, 4 = performance.

**Table 3 ijerph-19-06623-t003:** Factor structure matrix for stress coping.

Items	1	2
I try to do something to relieve stress	**0.919**	−0.088
I take active steps to change stressful situations	**0.889**	−0.044
I try to look at the stressful situation from a different perspective	**0.809**	−0.069
I think the current stress will be resolved soon	**0.702**	−0.025
I used to be in denial and think that the current stressful situation is not true	0.044	**0.859**
I do not want to believe that the stressful situation actually happened	0.209	**0.843**
I do not try to solve my stress	−0.269	**0.027**
I give up attempting to cope with my stress	−0.278	**0.789**
Eigenvalues	3.307	2.44
Variance (%)	41.34	30.506
Cronbach’s alpha	0.852	0.853

Note: 1 = active stress coping, 2 = passive stress coping.

**Table 4 ijerph-19-06623-t004:** Results of MANOVA.

Variables	Sub-Factors	*df*	*F*	*p ^a^*	*η^2^*
Stress	Fellow players	1	7.423	0.007 **	0.031
	Financial	1	2.622	0.107	0.011
	Family and coaches	1	2.159	0.143	0.009
	Performance	1	5.931	0.016 *	0.025
Stress coping	Active	1	1.156	0.283	0.005
	Passive	1	6.533	0.011 *	0.027
Quality of life		1	0.328	0.567	0.001

Note: *^a^* ** *p* < 0.01, * *p* < 0.05.

**Table 5 ijerph-19-06623-t005:** Mean scores of the dependent variables for the two groups.

	1	2	3	4	5	6	7
Group 1	**3.27 (0.89)**	2.89 (1.11)	2.77 (0.96)	**3.87 (0.82)**	3.64 (0.78)	**2.36 (0.75)**	3.48 (0.83)
Group 2	2.93 (1.03)	3.14 (1.16)	2.58 (1.00)	3.56 (1.09)	3.75 (0.76)	2.10 (0.82)	3.42 (0.81)

Note: Group 1 = KLPGA Tour players, Group 2 = Dream Tour players; 1 = fellow players, 2 = financial, 3 = family and coaches, 4 = performance, 5 = active, 6 = passive, 7 = quality of life. Statistically significant higher mean scores among groups are shown in bold. Standard deviations are shown in parentheses.

## Data Availability

The data presented in this study are available on request from the corresponding author.

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
