# Peer review of "Differences in Stress, Stress-Coping Behavior, and Quality of Life Based on the Performance of Korean Ladies Professional Golf Association Tour Players"

_ijerph, 2022, doi:10.3390/ijerph19116623_

Round 1

Reviewer 1 Report

This paper compares the stress, stress-coping behavior, and quality of life of women golfers in Korea, who play at the elite level, and the developmental level. It's a simple study, and I think it's close to publishable. However,  I have some questions and comments to raise with the authors that I would like them to consider.

The presentation of results and the discussion is adequate. We see there are some significant differences between the golfers and different levels, including stress from other players, stress from performance and passive coping factors. I'm interested in one aspect doesn't show significant differences - the active coping behaviors. I know we tend not to discuss results that aren't significant, but this seems a notable finding. Do the authors have any thoughts on why this expected difference did not occur?  Is it something methodological as noted as a possibility in the limitations, or are athletes at both levels engaging in high levels of active coping (lines 218-219?). Can we discuss this somewhere?

On the objective performance measure of being on the KLPGA vs Dream Tour - I think there is enough logic to this to make it a viable measure. But I'm left wondering if there might not be important differences within groups. In both tours, there are presumably some players who are winning a great deal (and thus performing better), and some players who are not performing so well and barely hanging in. Is stress and coping of better players different? I’m not saying the groups are invalid or anything, but this might be worth consideration in the discussion.

More minor issues:

Line 39 – can we change to “some of the world’s most competitive players…”

line 121 needs a reference?

Reference 25 – is that a translated title? Seems off a bit.

Author Response

Reviewer 1

  1. The presentation of results and the discussion is adequate. We see there are some significant differences between the golfers and different levels, including stress from other players, stress from performance and passive coping factors. I'm interested in one aspect doesn't show significant differences - the active coping behaviors. I know we tend not to discuss results that aren't significant, but this seems a notable finding. Do the authors have any thoughts on why this expected difference did not occur?  Is it something methodological as noted as a possibility in the limitations, or are athletes at both levels engaging in high levels of active coping (lines 218-219?). Can we discuss this somewhere? → We totally understand your concern. However, as you mentioned, researchers tend not to discuss results that are not statistically significant. I also am cautious to discuss statistically unproven results.
  2. On the objective performance measure of being on the KLPGA vs Dream Tour - I think there is enough logic to this to make it a viable measure. But I'm left wondering if there might not be important differences within groups. In both tours, there are presumably some players who are winning a great deal (and thus performing better), and some players who are not performing so well and barely hanging in. Is stress and coping of better players different? I’m not saying the groups are invalid or anything, but this might be worth consideration in the discussion. → We understand your comment. The KLPGA tour and Dream tour are separate professional golf tour, but they are fundamentally strongly connected under the Korean Ladies Professional Golf Association (KLPGA tour is much higher division). Thus, regardless of each player’s short-term performance, the type of golf tour can be an independent variable showing the objective performance. Nevertheless, your recommendation could be a meaningful approach for professional athlete. So, we will consider the approach on future research.

More minor issues:

  • Line 39 – can we change to “some ofthe world’s most competitive players…” → Based on your comment, the minor issue has been revised.
  • Line 121 needs a reference? → Additional references have been added.
  • Reference 25 – is that a translated title? Seems off a bit. → It is the title from original article.

Reviewer 2 Report

Thanks for the opportunity to read this article.

After reading, I have serious concerns that I present below.

Introduction. The entire introduction is highly speculative. The introduction should be written based on scientific evidence.

I could not follow the rationale of the introduction from the beginning to the aim of the study.

Lines 27-28. What does it mean growing qualitatively?

Lines 29-31. The sentence about the players can be deleted.

Lines 36-37. This is highly speculative. Remove this sentence.

What is the "literature review"?

This should be part of the introduction.

The aim of the study should be removed from the material and methods section. It should be moved to the introduction section.

Line 249. It is not clear what factors the authors are talking about.

Line 271. Delete ""(Group 1)".

This paper needs a conclusion.

Lines 297-302. This sentence should move to the discussion section. It can be the first sentence of the discussion.

Lines 303-323. This should move to the end of the discussion section.

Author Response

Reviewer 2

  1. The entire introduction is highly speculative. The introduction should be written based on scientific evidence.

I could not follow the rationale of the introduction from the beginning to the aim of the study.

→ We totally understand your comment. The section of Introduction has been revised.

  1. Lines 27-28. What does it mean growing qualitatively? → The sentence has been revised.
  2. Lines 29-31. The sentence about the players can be deleted. → The sentence has been deleted.
  3. Lines 36-37. This is highly speculative. Remove this sentence. → The sentence has been removed.
  4. What is the "literature review"? This should be part of the introduction. → Based on your comment, the “literature review” has been deleted.
  5. The aim of the study should be removed from the material and methods section. It should be moved to the introduction section. → We understand your concern. The sentence has been moved to the introduction section.
  6. Line 249. It is not clear what factors the authors are talking about. → Additional information has been added.
  7. Line 271. Delete ""(Group 1)". → Deleted.
  8. This paper needs a conclusion. → Based on your comment, the conclusion section has been revised.
  9. Lines 297-302. This sentence should move to the discussion section. It can be the first sentence of the discussion. → Based on your comment, the sentence has been moved to the Discussion.
  10. Lines 303-323. This should move to the end of the discussion section. → Based on your comment, the sentence has been moved to the Discussion.

Reviewer 3 Report

  • You should combine Introduction and Literature review as one section. Move the content of Literature review to Introduction.
  • Line 143, :August 19 to 22”, should it be 19th to 22nd  August 2021?
  • Same comment about the date in line 144.
  • In line 141, author mentioned 128 KLPGA, then in line 143, author mentioned 125. Thus total is 253 participants. In line 149, 249 were collected, so how many was excluded from both groups? Need to explain clearly in those lines.
  • Please include your study design, sampling method in recruiting participants, and eligible criteria (inclusion and exclusion) in section 3.1.
  • That will be good if author can include the sample size determination for this study.
  • Please mention which institution’s ethics research committee approve the study.
  • Table 1, please provide mean and SD for age, career as golf athletes (years).
  • Line 165, the author should explain why 3 sub factors were modified and deleted? If modified, will the questionnaire still valid and reliable? Which one is modified, and which one is deleted should be clearly explain?
  • Line 167-170, the alpha should be Cronbach alpha right? please mention what is alpha. Does this alpha obtain after modified/deleted the factors?
  • Line 176, “finally applied after minor revisions”, did the author revised the questionnaire before used in the present study? And which item was revised or changed? How you ensure the questionnaire is still valid and reliable with this “minor revision”?
  • Line 180, Again minor revision was made on quality-of-life scale. How do you ensure the validity and reliability of the questionnaire is still maintained?
  • In “data analysis”, authors conducted two different studies by using one same sample. Validation and multivariate analysis by using same sample. I think this is not appropriate. Author should have used different sample for these two studies. Can author provide good justification for this matter? Please give reference to support your justification.
  • Line 229, please include degree of freedom for F-statistics. Please give exact p-value not p<0.05.
  • Table 4, F-statistics should have 2 df not 1 df. Please correct.
  • Line 229, why Wilks lambda? This is not mentioned in section “data analysis”. Is this global F-stat for overall MANOVA? Since Table 4 has reported the F-stat for comparison between groups on each study variables, do you still need this Wilk lambda and F-stat values?
  • I suggest including the mean and SD of each study variables (separately for each group) in Table 4.
  • Table 5, what is c,d,e……? I think you can include the mean scores in Table 4 and remove Table 5.
  • Conclusion is about answer your research’s objectives and questions. I think many can move under Discussion.

My biggest concern is that the authors used same sample for two major studies, one is validation study and another one is MANOVA testing group differences on stress, coping and QOL. I hope the authors can give a good justification and rationale about this matter in the revision. 

Thank you for inviting me for this review. 

Round 2

Reviewer 2 Report

The authors did a good job and substantially improved the article. However, it is clearly noted that the text has some aspects that make me maintain my reservations about its publication.
For example, the verb tense of the objective should be in the past and not in the present. The way the introduction is written does not demonstrate rigour. The sentence someone says at a press conference is not relevant ("Seri stated "I've been a golfer since childhood and tried my best to win the competition against top players. However, I lacked preparation for a normal life other than my life as a golfer, and it was always mentally difficult"). The paper must have an introduction based on science and not opinions or press conference phrases.

Reviewer 3 Report

When study involve human being (although it is only questionnaire survey) ethical approval should be obtained from author's institution. However, I leave this to the Journal / editor to decide. 
